

# Validation of reference genes for gene expression studies in tartary buckwheat (*Fagopyrum tataricum* Gaertn.) using quantitative real-time PCR

Chenglei Li[1], Haixia Zhao[1], Maofei Li[1], Panfeng Yao[1], Qingqing Li[1], Xuerong Zhao[1], Anhu Wang[2], Hui Chen[1], Zizhong Tang[1], Tongliang Bu[1] and Qi Wu[1]

[1] College of Life Science, Sichuan Agricultural University, Ya'an, Sichuan, China
[2] Xichang College, Xichang, Sichuan, China

## ABSTRACT

Quantitative real-time reverse transcriptase polymerase chain reaction is a sensitive technique for quantifying gene expression levels. By implementing three distinct algorithms (geNorm, normFinder and BestKeeper), we have validated the stability of the expression of seven candidate reference genes in tartary buckwheat, including *FtSAND*, *FtCACS*, *FtExpressed1*, *FtGAPDH*, *FtActin*, *FtEF-1a* and *FtH3*. In this study, the results indicated that *FtCACS* and *FtSAND* were the best reference genes for 'abiotic cotyledons', *FtExpressed1* and *FtEF-1α* were the best reference genes for aluminium treatment, *FtCACS* and *FtExpressed1* performed the best for the immature seed stage, *FtCACS* was best for the abiotic treatment, and *FtH3* appeared to be the most suitable reference gene for the abiotic treatment in hypocotyls and all samples in this study. In contrast, *FtActin* and *FtGAPDH* are unsuitable genes. Our findings offer additional stable reference genes for gene expression research on tartary buckwheat at the immature seed stage and under abiotic treatment.

## INTRODUCTION

Quantitative real-time reverse transcription polymerase chain reaction (qRT-PCR) has become the most prevalent quantification method used in assays of gene expression on account of its specificity, accuracy, efficiency and high sensitivity (*Jiang et al., 2014*). The data from qRT-PCR can be analysed using absolute or relative quantification. Absolute quantification yields an exact gene copy number by transforming quantification cycles (*Hruz et al., 2011*) into a standard curve. Relative quantification relies on internal control genes as references to present the qRT-PCR data of the target genes (*Ginzinger, 2002*). Relative quantification has become a more widely used method in gene expression assays because most researchers mainly focus on the analysis of differences in gene expression (*Wu et al., 2016*). To normalise qRT-PCR data, a reference gene is needed, and it should be universally valid, with a similar expression level across all

Corresponding author
Qi Wu, wuqi@sicau.edu.cn

feasible cells, tissue samples and experimental treatments (*Huang et al., 2014*). However, ideal reference genes do not actually exist (*Remans et al., 2014*). The selection of the appropriate reference gene is a critical step in controlling the variability of samples when using a sensitive qRT-PCR technique (*Tong et al., 2009*). If the reference gene is appropriate, the discrepancies that may exist in terms of the initial sample amount, RNA integrity, RNA recovery and the efficiency of cDNA synthesis will be eliminated. Statistical algorithms, such as geNorm (*Vandesompele et al., 2002*), normFinder (*Andersen, Jensen & Orntoft, 2004*) and BestKeeper (*Pfaffl et al., 2004*), have been developed to help to select appropriate reference genes.

Most studies about gene expression in bacteria (*Wu et al., 2017*), yeast (*Cankorur-Cetinkaya et al., 2012*) and mammals (*Terzi et al., 2010*) now include reference gene validation (*Chapman & Waldenstrom, 2015*). However, the validation of reference genes in plants has received very little attention, and housekeeping genes tend to be used as references without any appropriate validation (*Gutierrez et al., 2008*). In 2005, *Czechowski et al. (2005)* were the first to present a list of stably expressed Arabidopsis genes under a large range of experimental conditions, and the evidence clearly showed that several genes are expressed more stably than traditional reference genes. Since then, suitable reference genes for gene expression studies have been reported in higher plants such as flax (*Huis, Hawkins & Neutelings, 2010*), soybean (*Hu et al., 2009*), tomato (*Lovdal & Lillo, 2009*), *Pisum sativum* (*Die et al., 2010*), *Brachypodium distachyon* (*Hong et al., 2008*) and carrot (*Tian et al., 2015*). However, reference gene stability is not consistent across experimental conditions and plant species. Consequently, it is necessary to find additional reference genes for different conditions and species.

Tartary buckwheat (*Fagopyrum tataricum* Gaertn.) belongs to the Polygonaceae family (*Kim et al., 2013*), and as an important functional food material, it has a relatively high flavonoid content. Tartary buckwheat is capable of thriving in regions with poor soil or harsh climates (*Kim et al., 2009*). The results of genome sequencing also show that tartary buckwheat has a remarkable ability to cope with highly variable environmental stress, including drought, salinity, UV-B and cold (*Zhang et al., 2017*). The stress resistance of tartary buckwheat is mainly due to its abundant flavonoids (*Suzuki, Honda & Mukasa, 2005*). In particular, tartary buckwheat is a naturally aluminium (Al) tolerant species (*Wang et al., 2015*). Abiotic stresses and flavonoid metabolism regulate the expression of these genes in plants at both transcriptional and post-transcriptional levels. There are many reports on tartary buckwheat under various experimental conditions in which gene expression is normalised to a reference gene (*H3*) for semi-quantitative RT-PCR or qRT-PCR (*Bai et al., 2014*). However, to date, there is no systematic strategy to analyse tartary buckwheat reference genes at the immature seed stage or under abiotic stress.

In this work, we aimed to evaluate the potential use of different reference genes for internal normalisation to more accurately measure the expression level of genes of interest in tartary buckwheat. Seven candidate reference genes were selected, and the stability of their expression was assessed in tartary buckwheat at the immature seed stage and under different abiotic stress treatments. Evaluating the stability of the expression of candidate

reference genes depends on statistical analysis. Three different statistical software programs (geNorm, normFinder and BestKeeper) were used to calculate the variability of the expression of the candidate genes and determine which were the most suitable.

## MATERIALS AND METHODS

### Plant materials and treatments

Tartary buckwheat ('Xiqiao No. 2') seeds were grown in the field on a farm (XTBG; 29°59′ N, 102°59′E; 800 m elevation) at Sichuan Agricultural University, Ya'an, Sichuan, China (*Li et al., 2012*). Tissues, including roots, stems, leaves, flowers, immature seed 1 (seed formation started) and immature seed 2 (seeds in the milk) were collected at the immature seed stage (*Gupta et al., 2011*). The 7-day-old seedlings were stressed with saline or drought by adding 100 mM NaCl or 20% PEG 2000, respectively, to the medium. The 7-day-old seedlings were exposed in a chamber at 4 °C with a 16 h photoperiod for cold treatment (*Gao et al., 2016*). UV-B treatment was conducted under UV-B (302 nm, 0.1 mW/cm$^2$) in a chamber. After 0, 2, 4, 6, 12 and 24 h of treatment, all stressed seedlings were collected and separated into cotyledons and hypocotyls. For the Al treatment, the samples were processed according to a previous report (*Zhu et al., 2015*). Root tips (zero to two cm) and basal roots (two to four cm) were sampled under both −Al and +Al conditions. All samples were collected in two biological replicates, and RNA was extracted immediately.

### Total RNA isolation and cDNA synthesis

Total RNA was isolated from various samples with an RNAout 2.0 kit (Tiandz, Beijing, China) according to the manufacturer's instructions. To remove trace DNA from samples, total RNA extractions were treated with RNase free DNase I. The RNA integrity was detected using 2% agarose gels. A Bio-RAD smart spec$^{TM}$plus spectrophotometer was used to determine the RNA purity. cDNA was synthesised with a PrimeScript$^{TM}$ RT reagent kit and gDNA Eraser (Perfect Real Time; TaKaRa, Dalian, China).

### Selection of candidate reference genes and design of qRT-PCR primers

Potential homologues of the seven candidate genes were identified from the transcriptome sequencing data of tartary buckwheat ('Xiqiao No. 2') (*Yao et al., 2017*). Primers were designed using Primer Premier 5.0. All primers used in this research are listed in Table 1. The specificity of the amplification was assessed based on the presence of a single band of the expected size in a 1.5% agarose gel following electrophoresis and a single peak in the qRT-PCR melting curve.

### Quantitative real-time PCR

The qRT-PCR procedure was designed in accordance with MIQE guidelines (*Bustin et al., 2009*). qRT-PCR was executed in a CFX96 Real Time PCR system (Bio-Rad, Hercules, CA, USA) with a SYBR Premix EX *Taq* kit ( TaKaRa, Dalian, China) in a total reaction volume of 15 μL that included 10 μL of SYBR Green mix, primers at 0.5 μM each and one μL
**Table 1 Genes, primers and different features derived from qRT-PCR analysis.**

| Gene symbol | Gene name | Amplification length (bp) | Primer sequences | E% | TM °C |
|---|---|---|---|---|---|
| *FtSAND* | SAND family protein gene | 79 | GACCCCCTTGCAGACAAAGCATTGGCA TCTCGTTCTCAACGTCTTTTACCCACTGG | 98.9 | 81 |
| *FtCACS* | Clathrin adapter complex subunit family protein | 125 | AAGACAGTCAGTTTCGTGCCACCTGA TCCATGCGTGTTCTACCCAACTCCTT | 90.3 | 82.5 |
| *FtExpressed1* | Expressed protein of unknown function | 127 | AGGCCAGTTCCTGCTGAATGTAATGC TAGCCTGATCCAAACAAGCCTGGCAA | 90.9 | 83 |
| *FtH3* | Histones 3 | 158 | GAAATTCGCAAGTACCAGAAGAG CCAACAAGGTATGCCTCAGC | 109.3 | 85 |
| *FtGAPDH* | Glyceradehyde-3-phosphate debydrogenase gene | 155 | TGGAGCTGCTAAGGCTGTCG TGATAGCACTCTTGATGTCCTCGTA | 91.9 | 83 |
| *FtActin* | Actin 2 gene | 118 | GGAAGTATAGCGTCTGGATTGGC CACTTGCGGTGAACGATTGC | 93.1 | 82.5 |
| *FtEF-1α* | Elongation factor-1α gene | 108 | GCTGCTGAGATGAACAAGAGGTC CTCAAACTTCCACAACGCGAT | 91.2 | 82.5 |

of cDNA. The amplification conditions were as follows: 95 °C for 30 s and 40 cycles of 95 °C for 5 s and 60 °C for 20 s. A melting curve from 60 to 95 °C was used to verify the specificity of the PCR amplification. All genes were amplified from cDNA. 10-fold serial dilutions of cDNA samples from young leaves were used to establish the standard curves to calculate the amplification efficiency of each primer pair.

## Statistical analysis

Three software programs (geNorm v3.5, the Excel add in of normFinder v0.953 and BestKeeper v1) were used to analysis the stability of reference gene expression across all experimental sets (*Andersen, Jensen & Orntoft, 2004*; *Pfaffl et al., 2004*; *Vandesompele et al., 2002*). SPSS v17.0 was used to calculate the span of Cq values for each gene by drawing a box-whisker plot, and the expression levels of *FtSTAR* and *FtDFR* were showed by mean ± standard deviation (SD).

## RESULTS

### RNA solution and quality

A series of 58 samples from tartary buckwheat were divided into six different groups. 'Abiotic cotyledons' was composed of cotyledons from all stress-treated samples. 'Abiotic hypocotyls' included hypocotyls from all stress-treated samples. 'Abiotic total' was composed of 'abiotic cotyledons' and 'abiotic hypocotyls'. 'Al treatment' comprised all samples that were treated with Al. 'Immature seed stage' consisted of six tissues (roots, stems, leaves, flowers, immature seed 1 and immature seed 2) at the immature seed stage based on the state of the seed. Finally, 'total' included all the samples in this study. Protein and organic pollutants were isolated and removed from all samples via RNA extraction. The total RNAs, with $A_{260}/A_{280}$ ratios of 1.8–2.0, were reverse transcribed into cDNA as templates for qRT-PCR detection.

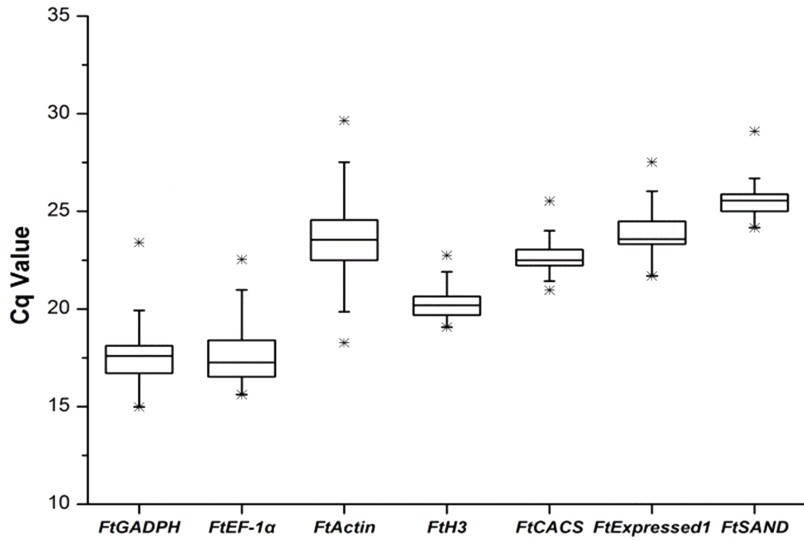

Figure 1 Cq values of seven candidate reference genes across all experimental sets in tartary buckwheat. A line across the box depicts the median. The box indicates the 25% and 75% percentiles. Whiskers represent the maximum and minimum values, and asterisks indicate extremes.

## Expression profiles of candidate reference genes

We selected the best reference genes (among the seven candidate genes) from the seven candidate genes in six different groups (sequencing data was shown in Fig. S1). The amplification efficiencies (E) of all reactions ranged from 95.5% to 107.5% and were calculated from standard curves with good linear relationships ($R^2 > 0.99$). Single-peak melting curves were obtained for all qRT-PCR amplifications (Fig. S2). A simple and commonly used method to identify stably expressed genes is to compare the span of quantification cycle (*Hruz et al., 2011*) values for each gene in the qRT-PCR reactions. The results showed that the candidate reference genes spanned a wide range of Cq values, ranging from 14.98 (*FtGAPDH*) to 29.65 (*FtActin*), with the median Cq values of the genes ranging from 17.59 (*FtGADPH*) to 25.5 (*FtSAND*). The reference gene expression levels, in descending order, were *FtGAPDH*, *FtEF-1*, *FtH3*, *FtCACS*, *FtActin*, *FtExpressed1* and *FtSAND*. *FtActin* (median, 23.55; Cq variation, 11.38) showed the most variability, and the *FtH3* gene (median, 20.19; Cq variation, 3.67) showed the least variability (Fig. 1).

## Software: geNorm, NormFinder and BestKeeper

After a simple comparison of the raw Cq values, the three software programs were used to further analyse the stability of the expression of the seven candidate genes. The genes were ranked in descending order in terms of the stability of their expression in each of six groups (Table 2), and the details of the cotyledons and hypocotyls under abiotic stress are provided in Table S1. GeNorm performed a stepwise exclusion of the most unstable gene and then recalculated *M* until only two genes remained, and these two genes had the most stable expression (*Vandesompele et al., 2002*). The gene with the lower *M*-value was considered to have the most stable expression. An *M*-value limit of <1.5 was

**Table 2 The stability of the expression of seven candidate reference genes in six groups as analysed by geNorm, NormFinder, BestKeeper.**

| Group | Rank | geNorm | | NormFinder | | BestKeeper | | |
|---|---|---|---|---|---|---|---|---|
| | | Gene | Stability | Gene | Stability | Gene | SD | CV |
| Abiotic Cotyledons | 1 | FtCACS | 0.33 | FtH3 | 0.058 | FtSAND | 0.33 | 1.32 |
| | 2 | FtH3 | 0.33 | FtCACS | 0.080 | FtCACS | 0.31 | 1.47 |
| | 3 | FtSAND | 0.41 | FtSAND | 0.129 | FtH3 | 0.33 | 1.64 |
| | 4 | FtExpressed1 | 0.54 | FtExpressed1 | 0.280 | FtExpressed1 | 0.59 | 2.48 |
| | 5 | FtEF-1α | 0.73 | FtEF-1α | 0.683 | FtEF-1α | 1.04 | 5.70 |
| | 6 | FtGAPDH | 0.84 | FtGAPDH | 0.751 | FtGAPDH | 1.07 | 6.26 |
| | 7 | FtActin | 1.31 | FtActin | 1.678 | FtActin | 1.46 | 6.36 |
| Abiotic Hypocotyls | 1 | FtH3 | 0.65 | FtSAND | 0.416 | FtH3 | 0.51 | 2.47 |
| | 2 | FtSAND | 0.65 | FtCACS | 0.437 | FtSAND | 0.69 | 2.68 |
| | 3 | FtCACS | 0.72 | FtH3 | 0.449 | FtActin | 0.79 | 3.38 |
| | 4 | FtExpressed1 | 0.84 | FtEF-1α | 0.469 | FtExpressed1 | 0.81 | 3.34 |
| | 5 | FtEF-1α | 0.91 | FtExpressed1 | 0.475 | FtCACS | 0.86 | 3.75 |
| | 6 | FtGAPDH | 1.05 | FtGAPDH | 0.811 | FtEF-1α | 0.92 | 5.32 |
| | 7 | FtActin | 1.17 | FtActin | 0.895 | FtGAPDH | 1.35 | 7.59 |
| Abiotic total | 1 | FtH3 | 0.55 | FtCACS | 0.304 | FtH3 | 0.52 | 2.55 |
| | 2 | FtSAND | 0.55 | FtSAND | 0.327 | FtCACS | 0.61 | 2.70 |
| | 3 | FtCACS | 0.61 | FtH3 | 0.349 | FtSAND | 0.63 | 2.46 |
| | 4 | FtExpressed1 | 0.72 | FtExpressed1 | 0.380 | FtExpressed1 | 0.73 | 3.02 |
| | 5 | FtEF-1α | 0.89 | FtEF-1α | 0.694 | FtEF-1α | 1.02 | 5.71 |
| | 6 | FtGAPDH | 1.01 | FtGAPDH | 0.782 | FtActin | 1.17 | 5.07 |
| | 7 | FtActin | 1.30 | FtActin | 1.329 | FtGAPDH | 1.20 | 6.89 |
| Al treatment | 1 | FtEF-1α | 0.16 | FtSAND | 0.091 | FtExpressed1 | 0.17 | 0.73 |
| | 2 | FtExpressed1 | 0.16 | FtExpressed1 | 0.100 | FtEF-1α | 0.25 | 1.55 |
| | 3 | FtSAND | 0.21 | FtH3 | 0.105 | FtH3 | 0.31 | 1.56 |
| | 4 | FtGAPDH | 0.23 | FtEF-1α | 0.106 | FtGAPDH | 0.34 | 1.88 |
| | 5 | FtH3 | 0.27 | FtGAPDH | 0.156 | FtCACS | 0.35 | 1.57 |
| | 6 | FtCACS | 0.30 | FtCACS | 0.282 | FtSAND | 0.37 | 1.43 |
| | 7 | FtActin | 0.49 | FtActin | 0.652 | FtActin | 1.05 | 4.16 |
| Immature seed stage | 1 | FtExpressed1 | 0.16 | FtSAND | 0.097 | FtExpressed1 | 0.50 | 2.00 |
| | 2 | FtCACS | 0.16 | FtGAPDH | 0.097 | FtCACS | 0.57 | 2.43 |
| | 3 | FtSAND | 0.30 | FtH3 | 0.167 | FtSAND | 0.82 | 3.03 |
| | 4 | FtGAPDH | 0.36 | FtCACS | 0.292 | FtEF-1α | 0.83 | 4.32 |
| | 5 | FtH3 | 0.45 | FtExpressed1 | 0.422 | FtGAPDH | 0.86 | 4.51 |
| | 6 | FtEF-1α | 0.62 | FtEF-1α | 0.675 | FtH3 | 1.11 | 5.29 |
| | 7 | FtActin | 1.01 | FtActin | 1.354 | FtActin | 2.16 | 9.96 |
| Total | 1 | FtH3 | 0.60 | FtCACS | 0.245 | FtCACS | 0.53 | 2.33 |
| | 2 | FtCACS | 0.60 | FtSAND | 0.304 | FtH3 | 0.54 | 2.69 |
| | 3 | FtSAND | 0.66 | FtH3 | 0.313 | FtSAND | 0.57 | 2.24 |
| | 4 | FtExpressed1 | 0.71 | FtExpressed1 | 0.377 | FtExpressed1 | 0.68 | 2.85 |
| | 5 | FtGAPDH | 0.87 | FtGAPDH | 0.650 | FtGAPDH | 0.97 | 5.53 |
| | 6 | FtEF-1α | 0.97 | FtEF-1α | 0.780 | FtEF-1α | 1.03 | 5.91 |
| | 7 | FtActin | 1.33 | FtActin | 1.480 | FtActin | 1.36 | 5.70 |

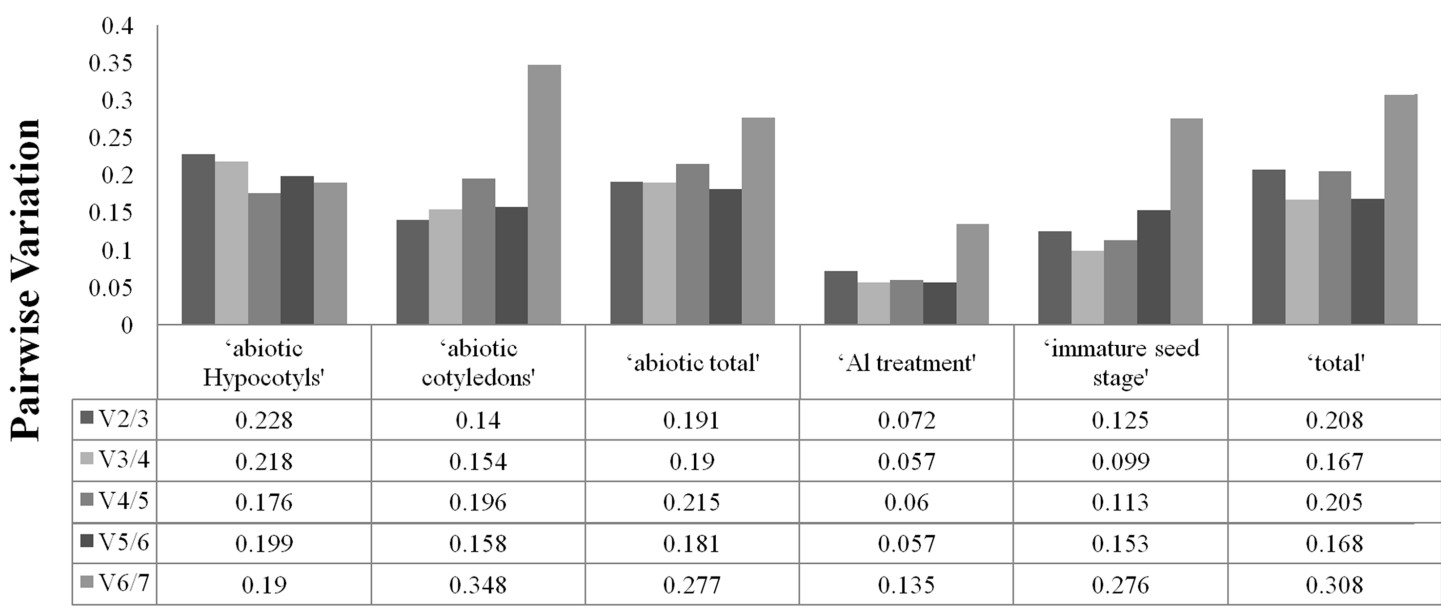

| | 'abiotic Hypocotyls' | 'abiotic cotyledons' | 'abiotic total' | 'Al treatment' | 'immature seed stage' | 'total' |
|---|---|---|---|---|---|---|
| ■ V2/3 | 0.228 | 0.14 | 0.191 | 0.072 | 0.125 | 0.208 |
| ■ V3/4 | 0.218 | 0.154 | 0.19 | 0.057 | 0.099 | 0.167 |
| ■ V4/5 | 0.176 | 0.196 | 0.215 | 0.06 | 0.113 | 0.205 |
| ■ V5/6 | 0.199 | 0.158 | 0.181 | 0.057 | 0.153 | 0.168 |
| ■ V6/7 | 0.19 | 0.348 | 0.277 | 0.135 | 0.276 | 0.308 |

**Figure 2 Optimal number of reference genes required for accurate normalization entile.**

suggested by geNorm. All the results from geNorm were lower than 1.5 in 'abiotic cotyledons' and 'total'; *FtCACS* and *FtH3* ($M$ = 0.33, 0.60) were chosen as the most stable genes, while *FtH3* and *FtSAND* ($M$ = 0.65, 0.55) were identified as the most stable genes in the 'abiotic hypocotyls' and 'abiotic total' groups, respectively, In 'immature seed stage', *FtCACS* and *FtExpressed1* ($M$ = 0.16) were recognised as the most stable genes. Finally, *FtEF-1α* and *FtExpressed1* ($M$ = 0.16) were the most stable genes in 'Al treatment'.

To determine the optimal number of reference genes required for accurate normalisation, geNorm was used to calculate the pairwise variation (Vn/Vn+1) between the sequential normalisation factors (NFs) (NFn and NFn+1). As suggested, a threshold value of 0.15 was adopted. As depicted in Fig. 2, pairwise variation analysis indicated that the ideal number of reference genes may be different for the different groups. For instance, only two genes are necessary for normalisation for 'immature seed stage', 'abiotic cotyledons' and 'Al treatment', but the pairwise variation of the other three groups was above the threshold of 0.15. The pairwise variations for cotyledons and hypocotyls under different abiotic treatments are provided in Fig. S3.

Unlike geNorm, normFinder (*Andersen, Jensen & Orntoft, 2004*) depends on a variance estimation approach, which allows the comparison of inter/intra-group variation. Genes with the lowest average expression stability values are the most stable. In the normFinder analysis, *FtH3* and *FtCACS* were the most stable genes in 'abiotic cotyledons', and *FtSAND* and *FtExpressed1* were the most stable genes under 'Al treatment'. In addition, the most stable reference genes in 'immature seed stage' were *FtSAND* and *FtGAPDH*, while the other three groups had the same two most stable reference genes (*FtSAND* and *FtCACS*). The least stable gene in all groups was *FtActin*.

BestKeeper (*Pfaffl et al., 2004*) determines the most stable genes by taking the coefficient of variance (CV) and SD of the Cq values. The more stably expressed genes are
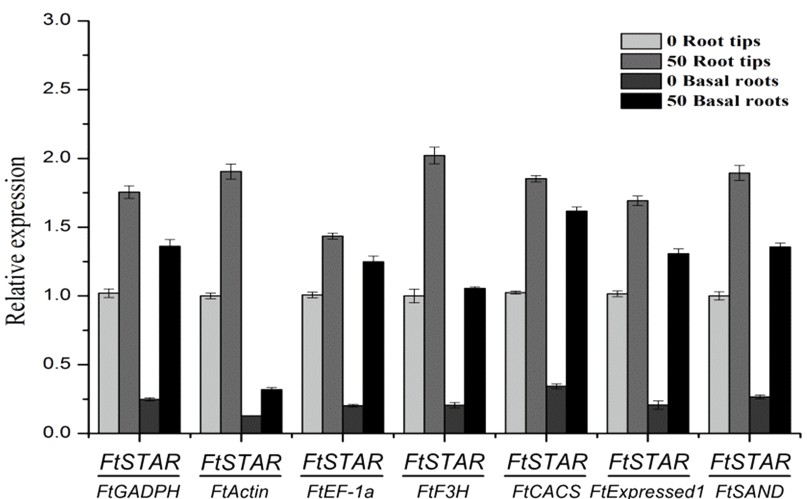

**Figure 3 Relative quantification of _FtSTAR2_ gene expression under Al treatment in tartary buckwheat.** Error bars represent standard deviation of the mean, data shown are means ± SD (_n_ = 3).

indicated by the lower SD and CV values. The results showed that the most stably expressed genes were _FtSAND_ (CV ± SD = 1.32 ± 0.33) and _FtCACS_ (CV ± SD = 1.47 ± 0.31) for 'abiotic cotyledons'. In 'abiotic hypocotyls', _FtH3_ (CV ± SD = 2.47 ± 0.51) and _FtSAND_ (CV ± SD = 2.68 ± 0.69) were the most stable genes. _FtH3_ (CV ± SD = 2.55 ± 0.52, CV ± SD = 2.69 ± 0.54) and _FtCACS_ (CV ± SD = 2.70 ± 0.61, CV ± SD = 2.33 ± 0.53) were the most stable genes in the 'abiotic total' and 'total' groups, respectively. _FtExpressed1_ (CV ± SD = 073 ± 0.17) and _FtEF-1α_ (CV ± SD = 1.55 ± 0.25) showed the most stable expression in 'Al treatment', and _FtExpressed1_ (CV ± SD = 2.00 ± 0.50) and _FtCACS_ (CV ± SD = 2.43 ± 0.57) showed the highest expression stabilities in 'immature seed stage'. The least stable gene was _FtActin_, but the _FtGAPDH_ gene, with a CV ± SD of 7.59 ± 1.39 in 'abiotic hypocotyls', was considered the least acceptable for gene expression normalisation.

## Reference gene validation

To validate the availability of a reference gene, the expression levels of _FtSTAR_ under Al treatment and of the _FtDFR_ gene under UV treatment were determined using the seven candidate reference genes for normalisation. For Al treatment, _STAR_ has a conserved response in plants, and this has been shown in previous reports, such as reports on rice _STAR2_ (_Huang et al., 2009_), _Arabidopsis ALS_ (_Larsen et al., 2005_) and tartary buckwheat _FtSTAR2_ (_Zhu et al., 2015_), whose expression increased after exposure to Al. The expression of _FtSTAR2_ was greatly increased (Fig. 3), which was reinforced by the Al-induced expression of _STAR2_. The use of the most favourable reference gene (_FtExpressed1_) resulted in the greatest variation, resulting in increases of approximately 1.67-fold in root tips and 6.24-fold in basal roots, and the use of the other most stable gene (_FtEF-1a_) resulted in increases of approximately 1.42-fold and 6.19-fold, respectively. Finally, the use of the least stable gene (_FtActin_) resulted in increases of

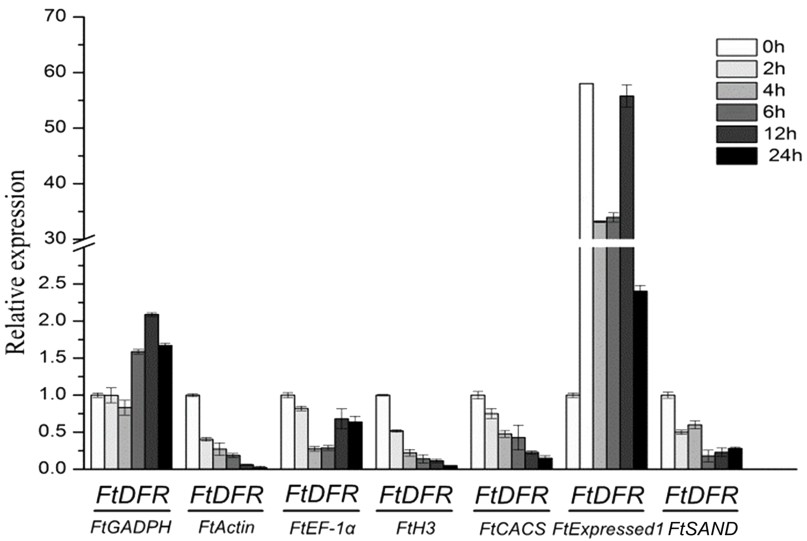

**Figure 4 Relative quantification of *FtDFR* gene expression under UV treatment in tartary buckwheat hypocotyls.** Error bars represent standard deviation of the mean, data shown are means ± SD ($n = 3$).

approximately 1.90-fold in root tips and 2.48-fold in basal roots. These genes also showed a pattern of increased expression, regardless of whether stable or unstable reference genes were used for normalisation. However, the relative expression using the most stable genes showed the higher fold increases compared with the unstable genes. Therefore, *FtExpressed1* and *FtEF-1a* are the most suitable genes under Al treatment.

Dihydroflavonol 4-reductase (DFR) catalyses the reduction of dihydroflavonols to leuco-anthocyanins and is a key enzyme in the biosynthesis of anthocyanins (*Yuan et al., 2007*). The contents of anthocyanin increased under UV treatment in tartary buckwheat hypocotyls (*Eguchi & Sato, 2009*). In hypocotyls under UV treatment, the relative quantification of *FtDFR* using the most stable genes (*FtCACS, FtH3* and *FtActin*) for normalisation exhibited similar expression patterns but different expression levels. When other less stable genes were used for normalisation, the results obviously differed from those with the most stable genes (Fig. 4). The expression levels of the *FtDFR* gene under UV treatment showed similar trends after normalisation to stable genes in hypocotyls. Thus, we selected the most stable reference genes for gene expression normalisation in tartary buckwheat.

## DISCUSSION

SYBR Green I is a fluorescent reporter dye used in qRT-PCR; when it binds double-stranded DNA, its fluorescence increases nearly 1,000-fold (*Morrison, Weis & Wittwer, 1998*). The problems in qRT-PCR caused by the variability of RNA templates and inappropriate data normalisation are obvious and widely known but disregarded (*Nolan, Hands & Bustin, 2006*). To address these questions, proper reference genes should be used. Thus, validating candidate reference genes and selecting stable reference genes are becoming general interests for researchers.
Candidate reference genes can be selected from reference gene validation papers that use the same experimental conditions or closely related species (orthologies) or by mining transcriptomic data for stably expressed genes or traditional housekeeping genes (*Hruz et al., 2011*). The seven candidate reference genes tested in tartary buckwheat include three genes that are stably expressed in common buckwheat (*SAND*, *CACS* and *Expressed1*), three commonly used reference genes (*GAPDH*, *Actin* and *EF-1a)* and one unique gene (*H3*). Neither high (>30) nor low (<15) Cq values are recommended based on general guidelines (*Wan et al., 2010*). In this study, the Cq values of the reference genes ranged from 14.98 (*FtGAPDH*) to 29.65 (*FtActin*), suggesting variable expression. Although the raw Cq value comparison can provide a rough estimate of the stability of gene expression, it is not sufficient to accurately evaluate the expression patterns of reference genes.

Scientists have developed several approaches to select stable reference genes for normalisation. However, to date, there is no consistent algorithm that can be used to test gene stability. These software programs have their own merits and drawbacks: NormFinder can avoid the misinterpretations caused by the artificial selection of co-regulated genes (*Andersen, Jensen & Orntoft, 2004*); geNorm is based on the assumption that none of the analysed genes are co-regulated (*Vandesompele et al., 2002*); and geNorm can estimate the fewest number of reference genes needed for accurate normalisation (*Stamova et al., 2009*). It seems possible that more reliable controls can be obtained using a combination of different algorithms (*De Almeida et al., 2010*). We used three software programs that were based on different statistical approaches to assess seven candidate genes in tartary buckwheat. The distinct statistical algorithms are likely to generate inconsistent rankings of stability. After integrating the outcomes of the three programs above in this study, we recommend the reference genes in tartary buckwheat as follows: *FtCACS* + *FtSAND* for 'abiotic cotyledons'; *FtExpressed1* + *FtEF-1α* for 'Al treatment'; *FtCACS* + *FtExpressed1* for 'immature seed stage'; *FtCACS* for 'abiotic total'; and *FtH3* for 'abiotic hypocotyls' and 'total'. The *H3* gene is the most commonly used internal gene for normalisation in tartary buckwheat (*Luo et al., 2016*). Our results indicate that *FtH3* is a stable gene under abiotic treatment. However, it is not stable across different organs or after Al treatment. In common buckwheat (*Demidenko, Logacheva & Penin, 2011*), *CACS* and *Expressed1* were validated as the most stable genes in the development and fruit stages. *FtCACS* and *FtExpressed1* being the most stable reference genes in the immature seed stage of tartary buckwheat supports the statement that the orthologues of identified reference genes could serve the same purpose in other species. In tomato, similar results were obtained in a study on reference gene selection (*Exposito-Rodriguez et al., 2008*).

*GAPDH* and *Actin* are the most generally used reference genes for the analysis of gene expression in various plant species (*Kumar et al., 2011*). However, our analysis indicates that *GAPDH* and *Actin* are not reliable genes for comparative expression analysis. The cause of these fluctuations at the gene expression level is probably that GAPDH and Actin have several biological functions, such as participating in the glycolytic pathway

and other processes (*Stürzenbaum & Kille, 2001*). Actin supports the cell and determines its shape, and it also takes part in other cellular functions (*Kravets, Yemets & Blume, 2017*). Although *Actin* is a stable reference gene in developmental studies, it is not stable under various conditions.

## CONCLUSIONS

As far as we know, our studies take advantage of three software (geNorm, NormFinder and BestKeeper) to analysis the stability of seven candidate reference genes for the first time in tartary buckwheat using qRT-PCR. The three software identified slightly different genes as most suited for normalisation, prompting us to merge the data. The result showed that the expression of *GAPDH* or *Actin* is unstable across all samples. We also provide a list with the stable reference genes in six group and under certain conditions.

### Funding
This work was supported by the National Natural Science Foundation of China (31500289). The funders had no role in study design, data collection and analysis, decision to publish, or preparation of the manuscript.

### Grant Disclosure
The following grant information was disclosed by the authors:
National Natural Science Foundation of China: 31500289.

### Competing Interests
The authors declare that they have no competing interests.

### Author Contributions
- Chenglei Li conceived and designed the experiments, performed the experiments, prepared figures and/or tables, approved the final draft.
- Haixia Zhao approved the final draft.
- Maofei Li conceived and designed the experiments, performed the experiments, prepared figures and/or tables, approved the final draft.
- Panfeng Yao approved the final draft.
- Qingqing Li approved the final draft.
- Xuerong Zhao approved the final draft.
- Anhu Wang contributed reagents/materials/analysis tools, approved the final draft.
- Hui Chen approved the final draft.
- Zizhong Tang approved the final draft.
- Tongliang Bu approved the final draft.
- Qi Wu conceived and designed the experiments, analyzed the data, authored or reviewed drafts of the paper, approved the final draft.
## Data Availability

The Genbank ID of reference sequences are MK416199 (*FtGAPDH*), MK430141 (*FtEF-1α*), MK416200 (*FtExpressed1*), MK416201 (*FtSAND*), MK430142 (*FtCACS*) and MK416202 (*FtSTAR2*).

## Supplemental Information

Supplemental information for this article can be found online at http://dx.doi.org/10.7717/peerj.6522#supplemental-information.

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
