# Peer review of "Validation of reference genes for gene expression studies in tartary buckwheat (Fagopyrum tataricum Gaertn.) using quantitative real-time PCR"

_PeerJ, doi:10.7717/peerj.6522_

## Round 0.1 · original submission · Minor Revisions

The three reviewers agreed that you manuscript is of good quality, requiring only several minor changes. It will be a useful resource for scientific community.

Reviewer 1 ·

Basic reporting

The manuscript by Li et al described the validation of 7 genes from Tartary buckwheat for use as reference genes in quantitative realtime PCR for gene expression study under abiotic stress condition. Good reference genes for different stresses or tissue types were reported. The study design was fine as it followed similar standard article of this kind. The manuscript was well-written with clear and plain English.

Experimental design

The study design was fine as it followed similar standard article of this kind.

Validity of the findings

The manuscript could attract reader's interest upon publication and may provide a set of reference genes for gene expression study in Tartary buckwheat in the future which may help easier to compare cross-study.

Additional comments

However, before the manuscript can be accepted for publication, this reviewer would like to ask the authors to address the followings:
The authors should provide a rationale on their selections of these 7 genes for validation. If a good rationale cannot be provided, the conclusion can only be restricted to "the best reference gene among the 7 genes validated", in some cases, there are best/better reference genes out there but were not included in this validation study. One of the way to check if the seven genes selected were actualy the best candidates is to survey in publicly available expression data of tatary buckwheat for the top stably expressed genes and see if these 7 genes were among the list.
As geNorm has been known to rank co-regulated genes as the best reference genes, the authors should address this in the discussion. What the authors will suggest to avoid this.
About the 7 genes selected for validation, the authors should link the gene names into respective gene IDs, if available. As we know especially in plant, there always several highly homologous members in a gene family. In the case gene IDs are not yet available, a full sequence of each of the candidate gene must be supplied for future reference of the work.
Some minor comments:
- Scientific name of Tartary buckwheat should be provided in the title

Reviewer 2 ·

Basic reporting

no comment

Experimental design

no comment

Validity of the findings

no comment

Additional comments

The authors present novel experimental study which results can be widely applied for gene expression studies in Fagopyrum. The manuscript is technically sound and I have only minor comments:

Lines 99-104 “Selection of candidate reference genes and design of qRT-PCR primers”
Could you provide here any rationale why exactly these genes were selected? Are the designed primers falls within single exon or not?

Line 117 - “Other statistical procedures were performed with SPSS v17.0. “ - please clarify what exactly was done;

Fig. 1 – please explain in the figure caption what the boxes and whiskers mean on the plot;

Fig. 3 and Fig. 4 - please explain in the figure caption what the whiskers mean;

Line 45 - add references for geNorm, normFinder and BestKeeper here (i.e. upon first mention).

Reviewer 3 ·

Basic reporting

- The manuscript "Validation of reference genes for gene expression studies in Tartary buckwheat using quantitative real-time PCR" described the validation of five ref genes for internal normalization to calculate the expression level of gene(s) in tartary buckwheat. The manuscript is suited to the PeerJ as per aims and scope.
- The Introduction (47 lines) has provided a comprehensive background of the necessary of ref gene(s) and the identification of the ref gene(s) in the kingdom.
- The figures and references were relevant and well-described.

Experimental design

- The materials and methods are suitable and were described carefully with right citations.

Validity of the findings

- The results and conclusion were fully stated and tightly linked with the research question.

Additional comments

- Comment #1: The authors should rephrase the sentence in L17-L20. The colon should not be used in the manuscript.
- Comment #2: The authors should check the tense in the sentence in L20-L24.
- Comment #3: Three statistical algorithms, including geNorm, normFinder and BestKeeper should be cited with right reference(s). (L45-46 and L.115-117).
- Comment #4: The sentence in L47-48 should be cited.
- Comment #5: Need a space between "in the work,we aimded" (L.70).
- Comment #6: FtEF-1α and FtSAND should be italicized in L.138.
- Comment #7: The author should abbreviate the quantification cycle (Cq) at the first mention (not like in L.175). The 'aluminum' should be used as 'Al' in whole text after the first mention of the abbreviation.
- Comment #8: "t" in "buckwheat" should not be italicized in L.192.
- Comment #9: The author should check the style of citation in L. 206.
- Comment #10: Please, check and put the style and word(s) of all references in the right form (REF. 4, 13, 15, 16, 17, 18, 19, 21, 22, 23, 24, 25, 27, 33, etc).
- Comment #11: The FtSTAR2 and FtDFR in the legend of Figure 3 and 4 should be italicized, respectively. All gene names in Figure 4 should be italicized.

---

## Round 0.2 · accepted · Accept

Dear authors

Thank you very much for submitting the revision on time. Your paper is now acceptable for publication. I hope that you will submit your next paper to PeerJ.

#